# Intrinsic Subtypes and Androgen Receptor Gene Expression in Primary Breast Cancer. A Meta-Analysis

**DOI:** 10.3390/biology10090834

**Published:** 2021-08-27

**Authors:** Paola Cruz-Tapias, Wilson Rubiano, Milena Rondón-Lagos, Victoria-E. Villegas, Nelson Rangel

**Affiliations:** 1School of Biological Sciences, Universidad Pedagógica y Tecnológica de Colombia, Tunja 150003, Colombia; pcruztapias@gmail.com (P.C.-T.); sandra.rondon01@uptc.edu.co (M.R.-L.); 2Hospital Universitario Mayor Méderi-Universidad del Rosario, 111411 Bogotá, Colombia; cirugia.seno@mederi.com.co; 3Centro de Investigaciones en Microbiología y Biotecnología-UR (CIMBIUR), Facultad de Ciencias Naturales, Universidad del Rosario, Bogotá 111221, Colombia; 4Departamento de Nutrición y Bioquímica, Facultad de Ciencias, Pontificia Universidad Javeriana, Bogotá 110231, Colombia

**Keywords:** androgen receptor, intrinsic subtypes, breast cancer, meta-analysis

## Abstract

**Simple Summary:**

Breast cancer (BC) is the second largest cause of death for cancer in women worldwide. Different studies have shown that the androgen receptor (AR), a cytoplasmic ligand-dependent transcription factor, may play a role as a marker of BC biology. We aimed to assess the clinical significance of *AR* gene expression in BC by meta-analysis of large-scale microarray transcriptomic datasets. Our findings suggest that high mRNA levels of *AR* have the potential to be a promising non-invasive prognostic biomarker for the identification of the less aggressive BC subtypes.

**Abstract:**

The androgen receptor (AR) is frequently expressed in breast cancer (BC), but its association with clinical and biological parameters of BC patients remains unclear. Here, we investigated the association of AR gene expression according to intrinsic BC subtypes by meta-analysis of large-scale microarray transcriptomic datasets. Sixty-two datasets including 10315 BC patients were used in the meta-analyses. Interestingly, *AR* mRNA level is significantly increased in patients categorized with less aggressive intrinsic molecular subtypes including, Luminal A compared to Basal-like (standardized mean difference, SMD: 2.12; 95% confidence interval, CI: 1.88 to 2.35; *p* < 0.001) or when comparing Luminal B to Basal-like (SMD: 1.53; CI: 1.33 to 1.72; *p* < 0.001). The same trend was observed when analyses were performed using immunohistochemistry-based surrogate subtypes. Consistently, the *AR* mRNA expression was higher in patients with low histological grade (*p* < 0.001). Furthermore, our data revealed higher levels of *AR* mRNA in BC patients expressing either estrogen or progesterone receptors (*p* < 0.001). Together, our findings indicate that high mRNA levels of *AR* are associated with BC subgroups with the less aggressive clinical features.

## 1. Introduction

Although more than 75% of all breast cancers (BC) are hormone receptor (HR) positive (Estrogen receptor-ER and/or Progesterone receptor-PgR), there is evidence indicating that estradiol and progesterone are not the unique hormones related to BC [1,2], suggesting the involvement of additional steroid molecules in BC, such as Androgen Receptor (AR). AR is expressed in 70–90% of mammary tumors [3] and it has been implicated in all stages of BC development [4]. However, the significance of its expression is not fully defined, since AR positivity (AR+) has been associated with different clinical outcomes and, contrary biological actions, all of them are apparently dependent on the ER status [5]. Thus, in ER positive (ER+) tumors, AR positivity is associated with reduced cell proliferation [6] and favorable clinicopathological features and prognoses [7], while in ER negative (ER-)/AR+ BC cases, it is generally accepted that AR promotes cell proliferation [8], although clinical studies have also reported contradictory results [9,10]. In this regard, clinical trials using enobosarm (selective AR modulator—SARM) or antiandrogens (i.e., bicalutamide and enzalutamide), when administered to ER+ BC patients, did not show clinical benefit rates (NCT02463032) [11,12]. However, a recent study has showed that in HR positive BCs, a subset of cases with both high levels of *AR* mRNA and low levels of *ESR1* (ER) mRNA, may benefit from enzalutamide [13]. This suggests that the utility of AR-targeted therapies may depend on the relationship between the levels of these HRs. By contrast, most of the clinical trials performed on ER- BCs have shown that antiandrogen treatment provides good results with considerable clinical benefit rates [14,15,16].

Since BC is considered a highly heterogeneous disease, it is usually classified in five intrinsic molecular subtypes (Luminal A, Luminal B, HER2-enriched, Basal-like, and Normal-like), determined by profiling mRNA expression of at least 50 genes (PAM50). Each one of these categories has different clinical and biological characteristics influencing patient prognoses [17]. However, in a clinical routine, subtype stratification is usually determined by immunohistochemical (IHC) expression of HR, human epidermal growth factor receptor 2 (HER2), and the proliferation marker Ki-67. This evaluation has allowed a surrogate-subtype classification to be established where cases are divided in Luminal A (HR+/HER2-/Ki-67 low), Luminal B/HER2-negative (HR+/HER2-/Ki-67 High), Luminal B/HER2-positive (HR+/HER2+/Ki-67 High), HER2-enriched (HR-/HER2+), and triple-negative BC—TNBC (HR-/HER-) [18]. The clinical classification has allowed researchers to group the great biological diversity that occurs in BC, in addition to determining prognoses and establishing more appropriate treatments for specific subtypes [19]. Despite, this it has been frequently reported that the BC subtype definition by both gene expression profiling as well as IHC shows discrepant results and they do not always identify the same lesions [20,21,22,23].

Although several works have evaluated the prognostic value of AR associated with HR status, few studies have directly evaluated the relationship between AR expression and BC subtypes. It has been shown that ER+/HER2- (Luminal A) tumors express AR (by IHC) more frequently than other IHC-surrogate subtypes and, in this BC subset, AR was considered an independent biomarker of positive outcome [3,24,25]. In contrast, AR prognostic value is less clear in TNBC, since its expression levels vary considerably in accordance with the high heterogeneity observed in this BC subtype [14,26]. On the other hand, preliminary gene expression analysis has revealed that AR mRNA levels are similar in all intrinsic molecular subtypes, except for the Basal-like subtype, where its levels are lower [27]. Moreover, it has been found that the prognostic value of AR expression seems to be dependent on the ER expression levels, and not only on the ER positivity and BC subtype determined, either by IHC or by molecular subtyping [28,29,30].

Considering the controversial data about the prognostic value of AR on BC subtypes, the present study aimed to investigate the association of AR gene expression with BC subtypes by conducting a meta-analysis of large-scale microarray transcriptomic datasets. In addition, to evaluate the associations of *AR* mRNA expression levels with different clinical and pathological characteristics, we established an agreement between *AR* gene expression levels with IHC-surrogate subtypes and with intrinsic molecular subtypes.

## 2. Materials and Methods

### 2.1. Microarray Databases Search Strategy in the Gene Expression Omnibus (GEO) Repository

Available microarray datasets related to BC were downloaded from the GEO repository in the National Center for Biotechnology Information (NCBI) website https://www.ncbi.nlm.nih.gov/geo/ (accessed on 20 May 2020). The final date for dataset inclusion was August 2020. The search strategy included the terms (“Breast Neoplasms” [Mesh]) AND “Expression Profiling” AND (“Homo sapiens” [porgn:_txid9606]).

The inclusion criteria were the following: (I) Enrolled data were obtained from humans; (II) Microarray datasets with information of *AR* gene expression; (III) The sample type is not cell lines; (IV) Enough information to calculate the standardized mean difference (SMD); (V) For clinical and pathological analyses, patients with primary invasive BC with clinical information available; (VI) BC patients without any treatment before the sample was taken.

### 2.2. Data Extraction

Based on the inclusion criteria, the following detailed clinical parameters were extracted: GEO accession number, PubMed identifier (PMID), BC histological type (invasive ductal carcinoma, invasive lobular carcinoma and mixed), tumor grade, nodal involvement, metastasis, ER status, PgR status, HER2 status, tumor size, and age. By following the model proposed by Carey et al. [31], IHC-surrogate subtypes were classified as Luminal A (HR+/HER2-), Luminal B (HR+/HER2+ and/or Ki-67 high), HER2-enriched (HR-/HER2+), or TNBC (HR-/HER2-). *AR* gene expression values were collected by using the tool GEO2R from NCBI. Microarray datasets using PAM50 [17], Hu et al. [32], or Sorlie et al. [33] algorithms, were included to establish the association among *AR* gene expression and intrinsic molecular subtypes (Luminal A, Luminal B HER2-enriched, Basal-like, and Normal-like).

### 2.3. Statistical Analysis

For each GEO dataset, the association between *AR* gene expression and BC was assessed by Student’s *t*-test or Mann–Whitney unpaired test based on normality distribution. The analyses were performed using the Statistical Package for Social Sciences (SPSS Version 25, Chicago, IL, USA).

For the meta-analysis, SMD with 95% confidence interval (95% CI) was used as a summary statistic because all studies measured the same outcome but at different scales. We conducted sensitivity analyses excluding outliers and, respectively, excluding studies with a small number (N) of participants. Outliers were defined as studies in which the pooled effect sizes 95% CI was outside the 95% CI of the pooled effect size (on both sides). We used an arbitrary cut-off of at least 20 randomized participants per arm for the analysis, excluding small N studies. Tough power calculations might differ from trial to trial, larger N trials are at least more precise in estimating the intervention effect [34].

Heterogeneity was calculated using Cochran’s (Q) and Higgins’ (I^2^) tests. The I^2^ test was expressed as a ratio ranging from 0% to 100%. The random-effects model was selected when I^2^ > 30% and *p*-value < 0.05. Otherwise, the fixed-effects model was selected. The presence of publication bias was graphically examined using funnel plots and Egger’s regression asymmetry tests. The Comprehensive Meta-Analysis version 3 program (Biostat, Englewood, NJ, USA 2004) was used for data analyses.

The degree of agreement between intrinsic molecular subtypes and IHC-surrogate subtypes was measured by Cohen’s kappa (κ). We considered the following levels of agreement: poor (κ < 0.0), slight (κ = 0.0–0.20), fair (κ = 0.21–0.40), moderate (κ = 0.41–0.60), substantial (κ = 0.61–0.80), and almost perfect (κ > 0.81) [35]. Statistical analyses were performed using SPSS v19 (IBM Corporation, Armonk, NY, USA).

## 3. Results

### 3.1. Association between AR mRNA Levels and Clinical-Pathological Characteristics of BC Patients

The initial search strategy identified 116,597 microarray datasets. After screening and eligibility assessment, a total of 62 datasets reporting *AR* mRNA expression levels in tumor specimens resected from BC patients (Figure 1) were included. The clinical and pathological characteristics evaluated are described in Table 1. Moreover, the description of the selected GEO studies is detailed in Appendix A.

Thirty-nine microarray datasets with clinical information of BC patients regarding their histological grade were included in the meta-analysis. Interestingly, in the primary analysis, the expression of *AR* was significantly increased in both tissues from patients with histological grade 1 (SMD: 0.427; 95% CI: 0.223–0.630; *p* < 0.001) and grade 2 (SMD: 0.408; 95% CI: 0.309–0.507; *p* < 0.001) compared to grade 3 tissues, with considerable heterogeneity (Appendix A). After exclusion of potential outliers, the trend of the pooled effect did not change and the expression of *AR* was still higher in both patients with histological grade 1 and grade 2 compared to patients with grade 3 (Figure 2A,B). Herein, the heterogeneity was moderate for both analyses. In these cases, there was no evidence of publication bias, based on the funnel plots and Egger’s test (Appendix A). There was no statistical evidence that tumor size, regional lymph node involvement, or the presence of distant metastasis had an association with *AR* gene expression levels.

From fifty microarray datasets, information about status of ER (7841 patients), PgR (4684 patients) and HER2 (4479 patients) was included in the primary analysis. The *AR* mRNA levels were significantly increased in women with ER+ BC (SMD: 0.634; 95% CI: 0.488–0.780; *p* < 0.001) (Appendix A) and in PgR+ BC patients (SMD: 0.576; 95% CI: 0.433–0.718; *p* < 0.001) (Appendix A). A similar increase was observed in ER+/PgR+ cases compared to ER+/PgR− cases (SMD: 0.187; 95% CI: 0.065–0.309; *p* = 0.003) (Appendix A). The heterogeneity was substantial for ER and PgR, and moderate for ER/PgR. After sensitivity analyses were conducted, we still observed a trend of the pooled effect towards a higher *AR* mRNA levels in BC patients expressing either ER (Figure 3A), PgR (Figure 3B) or being ER+/PgR+ (SMD: 0.210; 95% CI: 0.096–0.323; *p* < 0.001) (Appendix A). Here, the heterogeneity was moderate for ER and PgR, but low for ER/PgR.

Moreover, our results showed that *AR* mRNA levels were also increased in tissues from HER2+ BC patients regardless of their ER or PgR status (SMD: 0.174; 95% CI: 0.002–0.079; *p* < 0.001), with moderate heterogeneity (Appendix A). After exclusion of potential outliers, the trend of the pooled effect did not change and the expression of *AR* was still higher in HER2+ BC patients (Figure 3C).

Again, in none of these cases, evidence of publication bias based on the funnel plots and Egger’s test was observed neither for primary analyses (Appendix A) nor after sensitivity analyses (Appendix A).

### 3.2. Expression of AR Gene Is Increased in Luminal Subtypes

Since gene-expression profiling has had a considerable impact on the classification of BC, we next investigated whether the expression of *AR* is associated with the intrinsic molecular subtypes. In the primary analysis, fifteen microarray datasets with clinical information of BC patients classified as Luminal A (1168 cases), Luminal B (643 cases), HER2-enriched (359 cases), Basal-like (589 cases), and Normal-like (223 cases) were included in the meta-analysis. Our results showed that *AR* mRNA levels were higher in tumor tissues from patients classified as Luminal A, Luminal B, HER2-enriched, and Normal-like subtypes, when each one of them was compared to tissues from Basal-like cases (*p* < 0.001). Interestingly, after performing sensitivity analyses (Figure 4A–C), the findings were consistent with those from the primary analyses (Appendix A) and led to similar conclusions. Furthermore, the same association was found when comparing Luminal A, Luminal B, and HER2-enriched subtypes to Normal-like subtype (*p* < 0.001) (Table 2). In addition, *AR* gene expression was higher in Luminal A than in Luminal B (*p* < 0.001). However, there were no significant differences in *AR* mRNA levels when comparing Luminal A and Luminal B subtypes to HER2-enriched subtype (Table 2). Analyses of heterogeneity, as well as publication bias based on the funnel plots and Egger’s test, are presented in Appendix A.

Furthermore, given the clinical relevance of BC subtype classification, we also evaluated the association of *AR* expression with the IHC-surrogate subtypes. In the primary analysis, thirteen microarray datasets, including luminal A (1232 cases), luminal B (170 cases), HER2-enriched (199 cases), and TNBC (530 cases) patients were included in this meta-analysis. Herein, *AR* mRNA expression was significantly higher in tumor tissues from patients classified as Luminal A, Luminal B, and HER2-enriched compared to tissues from TNBC cases (*p* < 0.001). When the pooled effect was evaluated in sensitivity analyses (Figure 5A–C), the conclusions were consistent with those of the primary analyses (Appendix A). Furthermore, *AR* mRNA levels were higher in Luminal A than in HER2-enriched tumors (*p* = 0.02), but, in contrast, there were no significant differences in *AR* mRNA levels when comparing Luminal B to HER2-enriched subtypes (Table 2). Analyses of heterogeneity are detailed in Appendix A. Moreover, results concerning publication bias based on the funnel plot and Egger’s test are presented in Appendix A, as well as in Appendix A.

Finally, as we observed some differences between the results from intrinsic molecular subtypes and IHC-surrogate subtypes and their association with *AR* gene expression, we have provided an overview of the agreement between them. We included 11 studies in this analysis and, based on Cohen’s kappa (k), the agreement between Basal-like (molecular) and TNBC (IHC-surrogate) subtypes was predominantly substantial. Furthermore, the agreement of both classification systems was fair to moderate when analyzing Luminal A and HER2-enriched subtypes, but a slight concordance was observed when analyzing Luminal B tumors between molecular and IHC-surrogate subtype classification (Figure 6).

## 4. Discussion

In this study, we analyzed sixty-two published microarray gene transcriptomic datasets and found that, in BC patients, an increase in *AR* mRNA levels is associated with a low histological grade as well as the Luminal A subtype, defined by molecular or IHC-surrogate subtyping. These findings suggest that higher levels of *AR* mRNA may be related to BC tumors having less aggressive clinical features and good biological behavior.

The AR has gained increasing attention as a potential biomarker of BC [36]. Our analyses revealed higher *AR* mRNA levels in BC patients with a lower histological grade, which has been reported in previous gene expression analyses [37]. Since lower tumor grades are associated with cells that are slower-growing and look well-differentiated, such as the normal breast tissue, it is logical to assume that high *AR* gene expression levels may also be indicative of less aggressive BCs. Regarding other clinical characteristics, we observed no statistically significant correlation between tumor size, Ki-67 levels, nodal involvement, or distant metastasis with *AR* gene expression. However, previous studies have reported associations between AR protein expression by IHC with smaller tumor size and lower proliferative index (Ki-67 level) [38,39,40].

Nevertheless, AR significance in BC is still unclear, since AR positivity has been associated with different clinical outcomes in BC patients, according to the ER status. In ER+ BC, AR positivity is considered an independent prognostic factor of a good outcome, but in the subset of ER- BCs, there have been contradictory reports [5,9,10,41]. Here, the analyses performed in more than seven thousand BC patients showed that *AR* mRNA levels were significantly higher in women with ER+ tumors. This is consistent with previous reports studying *AR* gene expression levels [37,42]. The better outcomes observed in ER+ BCs having high AR expression may be attributed to the ability of AR signaling to consistently inhibit the basal and estrogen-induced proliferation and survival of ER+ BC cell lines [43,44,45]. AR has been suggested to antagonize ER signaling by competing with ER for binding to estrogen response elements (EREs) [46]. Furthermore, high *AR* mRNA levels were also maintained in PgR+ patients and even in a subgroup of ER+/PgR+ BC cases compared to the ER+/PgR− subgroup. These results are in agreement with those reported by Tagliaferri et al., who have indicated that low AR levels in cases with ER+/PgR− BCs may contribute to the identification of subgroups of high-risk patients [47]. In general, our results suggest that a global increase in *AR* gene expression seems to be a hallmark of HR+ BC.

The AR clinical significance was also studied according to intrinsic BC molecular subtypes. In agreement with what was observed in HR+, *AR* gene expression was significantly increased in Luminal A compared with other molecular subtypes. It is well established that BC molecular subtypes have unique prognoses and differ in their responsiveness to chemoprevention and chemotherapy [48]. Regarding prognosis, the Luminal A subtype has been shown to have a better outcome than the other subtypes across many datasets of patients with BC [19,49]. Thus, our data suggest that higher levels of *AR* mRNA may be associated with BC subtypes reported to be less aggressive and for having better prognosis, which is consistent with recent findings providing evidence that AR has a tumor suppressor role in ER+ BC [50]. These findings were confirmed with analyses performed using IHC-surrogate subtyping, except that, in this type of classification, differences among Luminal A and B subtypes were not significant, as it was observed with intrinsic BC molecular subtyping. Although Luminal B tumors have poorer outcomes and some of them can be identified by their expression of HER2, its major biological distinction is proliferation, which is higher in Luminal B than in Luminal A tumors [51]. Accordingly, some of the datasets used to determine IHC-surrogate subtyping did not include information about the proliferation marker Ki-67, so Luminal B tumors were defined as HR+/HER2+, following the model proposed by Carey et al. [31].

Additionally, our results showed a significant increase in *AR* gene expression in HER2+ BC patients. These patients are frequently associated with a poorer prognosis compared to HER2− BC cases [52]. Functional crosstalk between AR and HER2 have been described, which indicates that AR may cause a rapid initiation of cytoplasmic signaling cascades through the activation of the ErbB (HER family) and MAPK signaling in BC cells (non-genomic mechanism) [53,54]. Accordingly, high *AR* mRNA levels observed in the datasets studied here give support to the insight that, in HER2+ tumors, the cooperation between AR and HER2 promotes ERK activation that regulates both HER2 and AR gene expression. As a consequence, there is a positive feedback loop [55,56] that may stimulate cell proliferation and the worse clinical outcomes usually observed in HER2+ BCs. *AR* mRNA expression in these cases is consistent with analyses performed using both BC molecular subtypes and IHC-surrogate subtypes, showing that *AR* mRNA levels are higher in HER2-enriched tumors compared to basal-like and TNBC tumors, respectively. Remarkably, molecular and surrogate subtype stratifications did not show significant differences concerning *AR* mRNA expression levels between Luminal B and HER2-enriched tumors. This indicates the need to evaluate *AR* expression levels in combination with ER and HER2 to better characterize these BC subtypes. It has been well established that, in contrast to what has been observed in ER-/HER2+/AR+ tumors, HER2+ BC cases expressing high ER and AR levels have smaller tumor sizes, lower Ki-67 percentages, less aggressive phenotypes, and better outcomes [28,29,57,58].

Finally, kappa statistic was used to analyze the agreement between intrinsic molecular and IHC-surrogate subtype classification. Most of the studies included in the analyses showed a substantial agreement between basal-like (molecular) and TNBC (IHC-surrogate) subtypes; however, the agreement was fair for Luminal A and slight for Luminal B subtypes in many of them. This low agreement may further explain the differences observed in the association between *AR* gene expression levels and BC Luminal subtype classification systems. In addition, to assess several genes related to different biological processes, algorithms used to establish intrinsic molecular subtypes have a strong component based on the expression of genes associated with cell proliferation [17,32,33], that, as mentioned before, might not be included when IHC-surrogate subtyping is performed. Some studies have shown that the compatibility between the molecular and IHC-surrogate subtyping is still modest, with ranges of discordance between 15 and 19% for Luminal subtypes, while many of the tumors categorized as HER2-enriched by the molecular tests are not HER2+ by IHC, nor do they have *ERBB2* gene amplification [59,60,61]. This data indicates that BC subtypes defined by IHC must be carefully used to determine the potential of new BC biomarkers, since the information provided by this methodology might not be enough to replace molecular subtype classification [21,22], especially when several factors associated with IHC assessment can modify the detection and levels of the markers studied [62,63].

Our study had some limitations. First, it was not possible to determine the discriminative yield of AR expression for both disease-free survival and overall survival of BC patients. Consequently, survival analyses using the Kaplan–Meier estimator could not be conducted and does not allow us to be conclusive regarding BC prognosis. Moreover, for some analyses, there was substantial heterogeneity (I^2^ > 75%). However, after performing sensitivity analyses, and after potential outliers were excluded, residual heterogeneity decreased to between low and moderate. Limitations could be attributed to the differences in the number of participants between studies and because the measurement of gene expression levels may vary depending on sample processing or the type of microarray. It is very important to emphasize that the trend of the pooled effect observed before the application of sensitivity analyses did not change even after outliers were excluded. To address these limitations, future studies need to include larger cohorts of patients, and analyses may be performed with data from more standardized methodologies such as high throughput sequencing.

## 5. Conclusions

Here, we tested the association of *AR* mRNA levels in BC patients with intrinsic BC subtypes by conducting a meta-analysis of large-scale microarray transcriptomic datasets. Our analyses revealed higher levels of *AR* mRNA in BC cases expressing either ER and PgR, having a lower histological grade, and, in most cases, being categorized as Luminal A, the intrinsic molecular subtype characterized by good prognosis. The same trend was observed when the BC patients were classified using IHC-based surrogate subtypes. Our findings suggest that the analysis of mRNA levels of *AR* has the potential to be a promising non-invasive biomarker for the identification of the less aggressive BC subtypes. In line with these results, it will be interesting to identify luminal tumors in a clinical routine, having higher AR levels with respect to ER levels. This could be considered as an important medicinal target, since some clinical trials using antiandrogen therapies have reported significant clinical benefits for ER+ BC patients with high AR levels.

## Figures and Tables

**Figure 1 biology-10-00834-f001:**
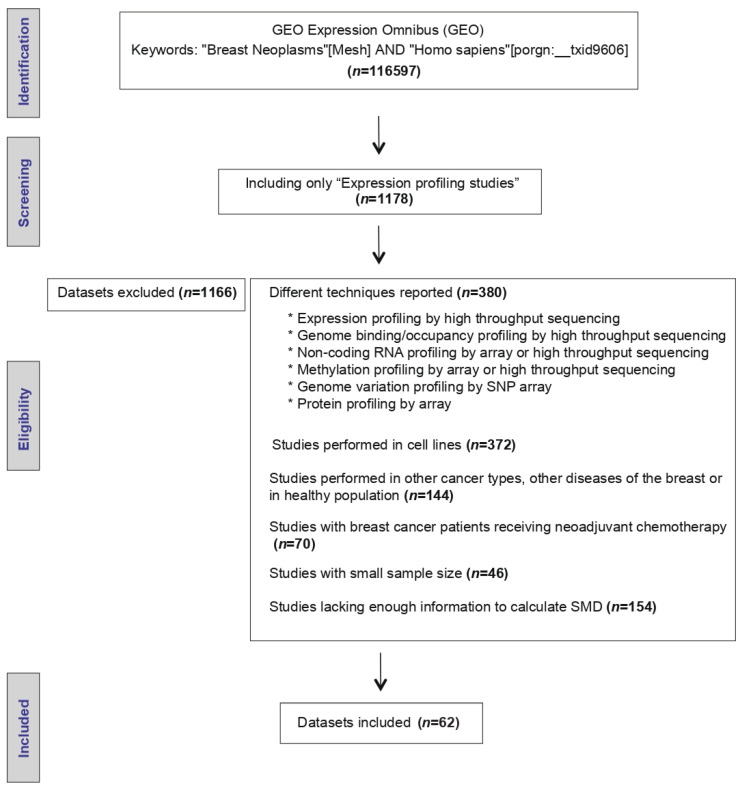
Workflow of the included Gene Expression Omnibus (GEO) datasets.

**Figure 2 biology-10-00834-f002:**
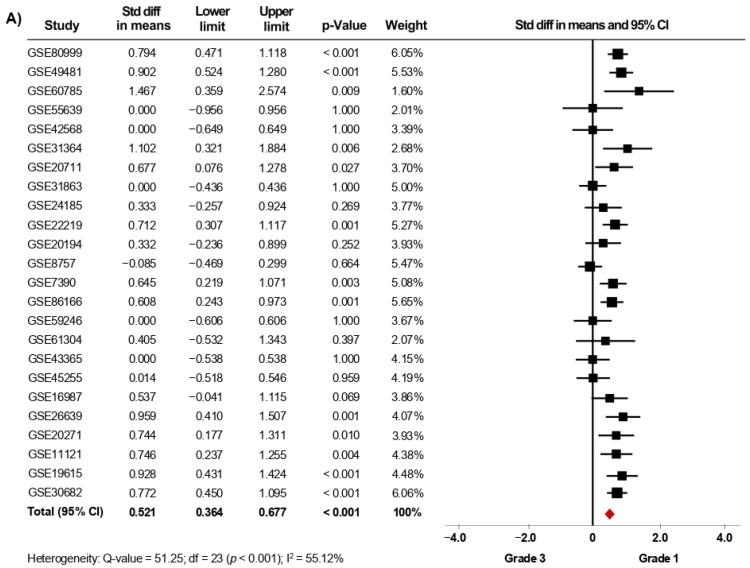
*AR* mRNA expression is significantly increased in BC patients with low histological grade. Results after sensitivity analyses. Forest plots of SMD comparing *AR* mRNA levels in BC patients with histological grade 3 vs. BC patients with histological grade 1 (**A**), and grade 2 (**B**). The squares represent the SMD for each dataset. The horizontal line crossing the square represents the 95% CI. The red diamonds represent the estimated overall effect.

**Figure 3 biology-10-00834-f003:**
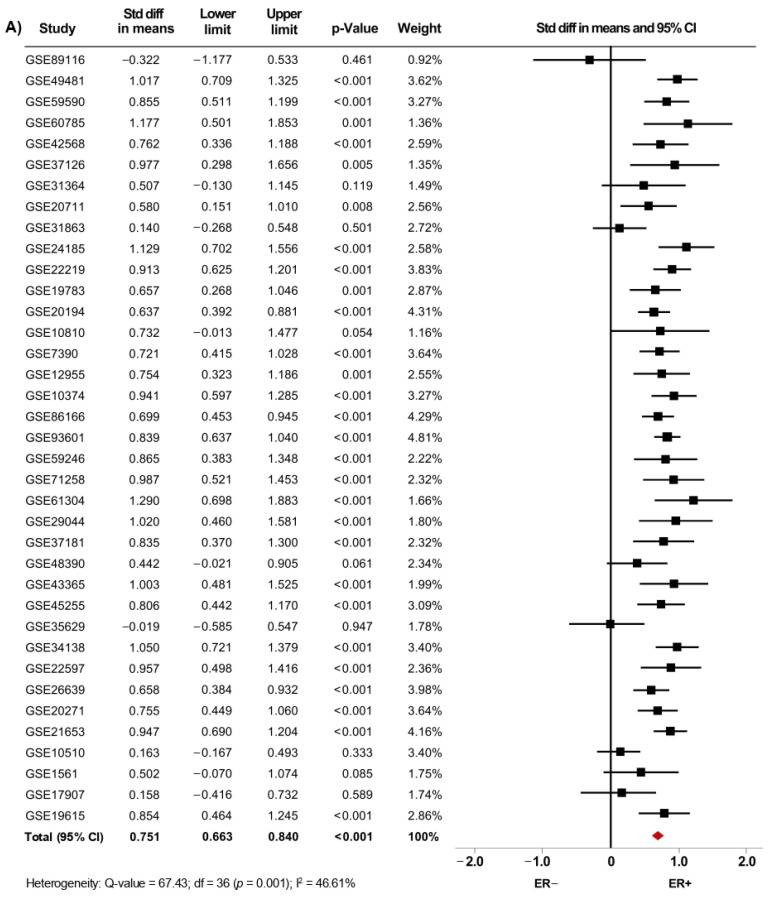
*AR* mRNA expression is higher in BC patients expressing either ER, PgR and HER2. Results after sensitivity analyses. Forest plots of SMD showing *AR* mRNA levels associated with ER (**A**), PgR (**B**), and HER2 status (**C**). The squares represent the SMD for each dataset. The horizontal line crossing the square represents the 95% CI. The red diamonds represent the estimated overall effect.

**Figure 4 biology-10-00834-f004:**
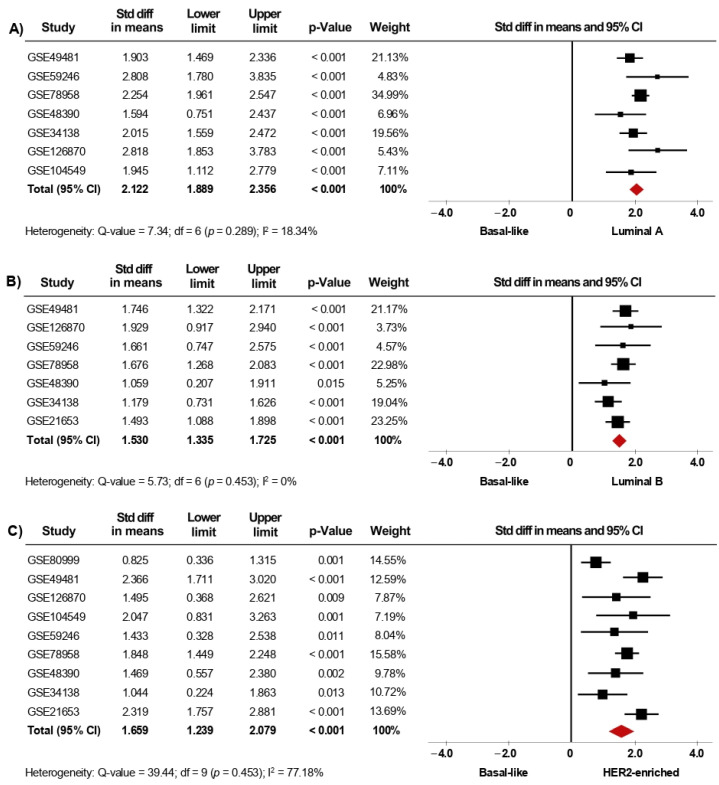
*AR* mRNA level is significantly increased in patients categorized within the less aggressive intrinsic molecular subtypes. Results after sensitivity analyses. Forest plots of SMD comparing *AR* mRNA levels in BC patients classified as Basal-like subtype vs. Luminal A (**A**), Luminal B (**B**), and HER2-enriched subtypes (**C**). The squares represent the SMD for each dataset. The horizontal line crossing the square represents the 95% CI. The red diamonds represent the estimated overall effect.

**Figure 5 biology-10-00834-f005:**
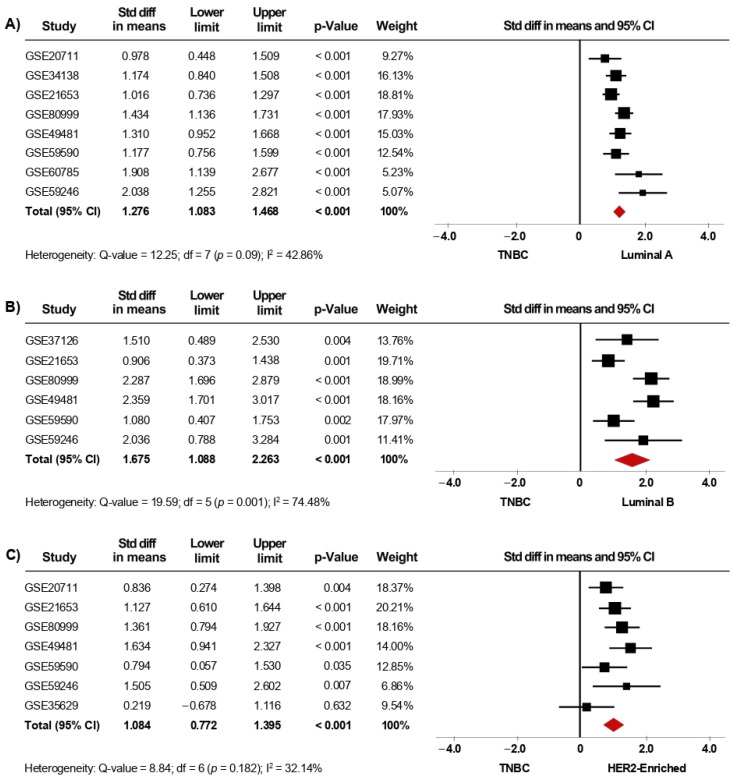
*AR* mRNA level is significantly increased in patients categorized with less aggressive IHC-surrogate subtypes. Results after sensitivity analyses. Forest plots of SMD comparing *AR* mRNA levels in BC patients classified as TNBC subtype vs. Luminal A (**A**), Luminal B (**B**), and HER2-enriched subtypes (**C**). The squares represent the SMD for each dataset. The horizontal line crossing the square represents the 95% CI. The red diamonds represent the estimated overall effect.

**Figure 6 biology-10-00834-f006:**
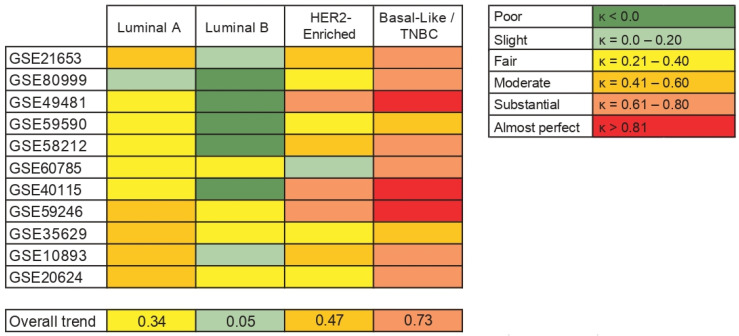
Agreement between intrinsic molecular subtypes and IHC-surrogate subtypes in terms of its association with *AR* gene expression. The table on the left shows the levels of agreement in terms of the kappa coefficient (k).

**Table 1 biology-10-00834-t001:** Clinical characteristics of BC patients.

Characteristics		*n* (%)
Grading	1	801 (13.5)
	2	2399 (40.5)
	3	2719 (45.9)
Regional lymph nodes (N)	N0	1899 (55.2)
	N1	1537 (44.7)
Distant metastasis (M)	M0	971 (80.1)
	M1	241 (19.8)
Estrogen receptor (ER)	Positive	5171 (65.9)
	Negative	2670 (34.1)
Progesterone receptor (PgR)	Positive	2463 (52.5)
	Negative	2221 (47.4)
HER2	Positive	1028 (22.9)
	Negative	3451 (77.1)
IHC—Surrogate Subtype	Luminal A	1232 (57.8)
	Luminal B	170 (7.9)
	HER2-enriched	199 (9.3)
	TNBC	530 (24.8)
Intrinsic Molecular Subtype	Luminal A	1168 (39.1)
	Luminal B	643 (21.5)
	HER2-enriched	359 (12.0)
	Basal-like	589 (19.7)
	Normal-like	223 (7.5)

Abbreviations: Human epidermal growth factor receptor 2 (HER2), Immunohistochemistry (IHC), Triple Negative Breast Cancer (TNBC).

**Table 2 biology-10-00834-t002:** Analysis of the association between *AR* mRNA expression and BC classification based on molecular subtypes and IHC-surrogate subtypes after sensitivity analyses.

		Std Diff in Means	Lower Limit	Upper Limit	*p*-Value
Molecular subtypes	Basal-like vs. Luminal A	1.940	1.399	2.482	<0.001
	Basal-like vs. Luminal B	1.405	1.040	1.771	<0.001
	Basal-like vs. HER2-enriched	1.487	1.007	1.966	<0.001
	Basal-like vs. Normal-like	1.144	0.720	1.568	<0.001
	Normal-like vs. Luminal A	0.776	0.599	0.954	<0.001
	Normal-like vs. Luminal B	0.463	0.218	0.708	<0.001
	Normal-like vs. HER2-enriched	0.676	0.415	0.937	<0.001
	HER2-enriched vs. Luminal A	0.016	−0.116	0.148	0.812
	HER2-enriched vs. Luminal B	−0.143	−0.357	0.071	0.190
	Luminal B vs. Luminal A	0.435	0.296	0.575	<0.001
IHC-surrogate subtypes	TNBC vs. Luminal A	1.356	1.135	1.576	<0.001
	TNBC vs. Luminal B	1.436	0.748	2.124	<0.001
	TNBC vs. HER2-enriched	1.084	0.772	1.395	<0.001
	HER2-enriched vs. Luminal A	0.193	0.022	0.365	0.027
	HER2-enriched vs. Luminal B	0.083	−0.145	0.310	0.477
	Luminal B vs. Luminal A	0.081	−0.111	0.274	0.409

Abbreviations: Human epidermal growth factor receptor 2 (HER2), Immunohistochemistry (IHC), Triple Negative Breast Cancer (TNBC).

## Data Availability

Available microarray datasets related to BC were downloaded from the GEO repository in the NCBI website (https://www.ncbi.nlm.nih.gov/geo/ (accessed on 21 May 2021)). Datasets included in this study are listed in Appendix A.

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
