# Peer review of "Intrinsic Subtypes and Androgen Receptor Gene Expression in Primary Breast Cancer. A Meta-Analysis"

_biology, 2021, doi:10.3390/biology10090834_

Round 1

Reviewer 1 Report

A very interesting and well-performed metanalysis analizing 62 published microarray gene transcriptomic datasets. Analysis showed that an increase in androgen receptor mRNA levels may be associated with a low histological grade as well as Luminal A subtype, defined by molecular or immunohistochemical-surrogate subtyping in breast cancer patients. Androgen receptor mRNA levels, according to the author's metanalysis, may be related to less aggressive clinical breast cancers. The statistical analysis seems well performed and the results are significant. I would probably just expand the introduction paragraph stating different treatments already present for the management of breast cancer; also in the discussion/conclusion paragraph, I would probably focus more on the possible treatment considering the minor aggressiveness of AR breast cancer here an article you could consider: doi: 10.1007/s40264-021-01071-1. 

Also, the template seems the one for the journal "cancers", instead of the one for "biology"; authors should modify it.

Thank You

Reviewer 2 Report

The manuscript presents a systematic review and meta-analysis of AR mRNA expression across different breast cancer subtypes. The patients' data are stratified by histological grade (1-3), histological markers (ER/PgR/HER2), and intrinsic molecular subtypes. It was found that AR expression is associated with the less aggressive subtypes. It demonstrates the translational value of AR mRNA expression as a prognostic marker.

The authors present a creative use of existing transcriptomic/microarray data. This approach can be used in the rapid biomarker discovery in a wide range of cancers.

Minor comments:

  1. If the manuscript fulfils all items in the PRISMA checklist, the phrase "A systematic review and meta-analysis" can be added to the title. Articles of similar genres can be found in the journal.
  2. The fold changes in Figures 2 and 3 were mostly less than 2 (in linear scale). While they are statistically significant and important findings, it would be difficult to detect minor fold changes in a clinical setting.
  3. If possible, the AR expression trend can be validated with an independent tissue microarray. If the same trend is observed, the result would be even more conclusive.

Reviewer 3 Report

This is a meta-analysis mentoring the impacts of androgen receptor (AR) in the intrinsic subtype of breast cancer. The author aimed to investigate the association in between AR expression and intrinsic type. The author screened 1178 studies and enrolled 62 reports; first, results showed that the expression level of AR was significantly higher in ER(+), PR(+) and histologic low grade tumors. Second, expression of AR was much higher in Luminal A breast tumors, compared to that in other subtypes, especially in TNBC. Based on these findings, the author concluded that high mRNA levels of AR was a promising non-invasive prognostic biomarker for the identification of the less aggressive breast cancer subtypes.

Major concerns

The author concluded that the AR expression was a prognostic marker, and associated with a subgroup of better prognosis. Because the author found that AR(+) breast cancer was associated with ER(+) expression and low histologic grade, it was presumed that AR(+) breast cancer has a better prognosis. However, this manuscript did not meta-analyze the disease-free survival (DFS) or overall survival (OS) of patients with AR(+) breast cancer. Under NO direct evidence, this kind of conclusion is not adequate.

In previous studies, around 70-90% ER(+) breast cancers express AR, 30-60% Her2(+) breast cancers express AR and 10-20% breast cancers express AR. Thus, the mean mRNA expression of AR in ER(+) breast cancer is originally superior than the mean AR expression in TNBC. The finding from the author is compatible with previous results, and not a new finding. Whether AR predicts the prognosis remains controversial, and its prognostic value is variable among different subtypes. In the trial BIG 1-98, AR expression was not associated with prognosis in the ER(+) breast cancer. In several previous TNBC studies, AR(+) TNBC was associated with a better survival, compared to AR(-) TNBC.

Several studies enrolled in this manuscript reported the DFS or OS. I suggest that the author should meta-analyze the DFS or OS of AR(+) breast cancer among the overall breast cancer patients and among the each subtype. Then, the author can make a conclusion based on this result.

The important and unknown issues of AR are whether AR predicts survival? In all breast cancer or only in a subtype? For patients with AR(+) breast cancer, what’s role of the adjuvant chemotherapy and/or endocrine therapy? The impact of AR inhibition therapy (such as Bicalutamide, although several trials ongoing)?
